Enhanced detection of accounting fraud using a CNN-LSTM-Attention model optimized by Sparrow search

Wu Peifeng w_peifeng1999@163.com
Chen Yaqiang
School of Statistics, Jilin University of Finance and Economics , Changchun , Jilin , China
Do Trang
Electronic publication date: 2024 Nov 26
Publication date: 2024
Volume: 10
Electronic Location ID: e2532
Received 2024 Sep 27; Accepted 2024 Oct 29
Copyright: ©2024 Wu and Chen
Copyright year: 2024
Copyright holder: Wu and Chen
License: This is an open access article distributed under the terms of the Creative Commons Attribution License, which permits unrestricted use, distribution, reproduction and adaptation in any medium and for any purpose provided that it is properly attributed. For attribution, the original author(s), title, publication source (PeerJ Computer Science) and either DOI or URL of the article must be cited.
License URL: https://creativecommons.org/licenses/by/4.0/

Keywords: Accounting fraud, Prediction, Neural networks, Attention, Sparrow search

Funding: The authors received no funding for this work.

==============================
The detection of corporate accounting fraud is a critical challenge in the financial industry, where traditional models such as neural networks, logistic regression, and support vector machines often fall short in achieving high accuracy due to the complex and evolving nature of fraudulent activities. This paper proposes an enhanced approach to fraud detection by integrating convolutional neural networks (CNN) and long short-term memory (LSTM) networks, complemented by an attention mechanism to prioritize relevant features. To further improve the model’s performance, the sparrow search algorithm (SSA) is employed for parameter optimization, ensuring the best configuration of the CNN-LSTM-Attention framework. Experimental results demonstrate that the proposed model outperforms conventional methods across various evaluation metrics, offering superior accuracy and robustness in recognizing fraudulent patterns in corporate accounting data.

Introduction

Reasonable standardization of accounting information is essential for ensuring justice, fairness, and transparency in the securities market. However, as finance evolves and technology advances, accounting fraud in enterprises has become more prevalent, especially since the 1990s. This fraud has significantly impacted economic development. Zager, Malis & Novak (2016) highlighted that the most common fraudulent financial practices involve overstating assets and understating liabilities. They also raised concerns about corporate accountability and the prevention of financial fraud, emphasizing that such measures are often ineffective in curbing corporate misconduct and can be difficult to detect. Beasley et al. (2000) researched accounting fraud across various industries, noting that methods differ by sector: financial companies often engage in asset abuse, while technology companies typically inflate income. Such fraud undermines investor confidence, harms economic interests, disrupts securities market transactions, and introduces financial risk to the financial sector.

Identifying fraud is crucial for investor decision-making. For example, Albrecht & Romney (1986) conducted a sample survey of companies and identified variables as indicators of accounting fraud. Persons (2011) used 10 financial indicators as training features in a logistic stepwise regression algorithm to identify accounting fraud, finding significant effects from certain variables. Green & Choi (1997) applied a neural network to identify financial fraud, developing a screening method. Fanning & Cogger (1998a) compared logistic regression, quadratic discrimination, neural networks, and linear discrimination, concluding that neural networks are more accurate in identifying accounting fraud.

Lee, Ingram & Howard (1999) analyzed 56 fraudulent companies and their non-fraudulent counterparts from 1978 to 1991, establishing a logistic stepwise regression model. They found that the earnings cash flow difference index is significant in identifying fraud, with fraudulent companies showing higher differences. Bell & Carcello (2000) compared logistic regression and artificial neural networks in identifying fraud among 42 companies, finding that neural networks have high recognition accuracy. Lin, Hwang & Becker (2003) developed an feedforward neural network (FNN) accounting fraud discrimination model based on income index and trend indicators, validating its rationality. Cecchini et al. (2010) created a support vector machine model with an improved radial basis kernel function, achieving effective fraud detection across different years. Ozdagoglu et al. (2016) identified fraud in Turkish listed companies from 2009 to 2013 using decision trees, logistic regression, and neural network models, finding that neural networks performed the best. Fanning & Cogger (1998b) found that the random forest algorithm can be effectively used to detect and analyze financial fraud. They ranked the importance of various factors and identified three key indicators for random forest analysis: the cash asset ratio, equity multiplier, and capital accumulation rate.

Currently, in the field of corporate financial accounting fraud identification, the most commonly used models include neural networks, logistic regression, and support vector machines. However, the accuracy of these models in detecting corporate accounting fraud is often low. This paper presents a comprehensive approach to enhancing the recognition accuracy of corporate accounting fraud by integrating convolutional neural networks (CNN) and long short-term memory (LSTM) networks, incorporating an attention mechanism, and optimizing parameters using the sparrow search algorithm (SSA). Deep learning enables businesses to automatically extract insights from vast amounts of complex and unstructured data, such as financial transactions, customer behaviors, and market trends (Leow, Nguyen & Chua, 2021). By applying advanced architectures like CNN-LSTM with attention mechanisms, companies can uncover hidden patterns, predict market shifts, and detect anomalies more effectively. To demonstrate the effectiveness of our proposed method, experiments were conducted to analyze and compare its performance against CNN, LSTM, CNN-LSTM, CNN-LSTM-Attention, and Learning Vector Quantization (LVQ) neural networks across various evaluation metrics.

Materials and Methods

Data collection and preprocessing

The dataset used in this article is sourced from the CSMAR (Guotai’an) database, spanning the years 2014-2021. After matching the data, labels were assigned: a label of 0 indicates that a company did not commit financial fraud, while a label of 1 signifies that the company engaged in financial fraud during the specified year. The data was further refined by filtering relevant indicators and removing missing values and outliers to minimize the impact of data anomalies.

The data indicators in this article undergo dimensionality reduction using principal component analysis (PCA) combined with the Pearson correlation coefficient for collaborative processing. The primary goal of dimensionality reduction is to eliminate redundant and irrelevant information present in the high-dimensional original data, which can otherwise lead to reduced accuracy during the identification process.

PCA achieves dimensionality reduction by transforming the high-dimensional data into a lower-dimensional space through specific mapping. The core idea behind PCA is to preserve as much of the essential features from the original data as possible while maximizing the variance in the reduced dimensions, thereby minimizing redundant information.

The PCA process involves several steps. First, the sample data is centered by calculating the mean, followed by obtaining the covariance matrix. Then, the eigenvalues and eigenvectors of the covariance matrix are computed. The eigenvalues are sorted in descending order, and the corresponding top eigenvectors are retained. Finally, the original data is transformed into a new set of principal components, effectively reducing its dimensionality.

For dimensional vectors w in the target subspace n, the original data is transformed using the following mapping function to maximize variance: (1) maxw1m−1∑i=1mwTxi−x¯2,

where m represents the number of samples, xi is data sample ith, and x¯ is the average vector of all data samples. PCA reduces the original dimension X to a K dimension, which can be expressed as Y = WTX.

PCA dimensionality reduction optimizes the data structure to maximize the retention of essential information within the data’s internal characteristics. However, PCA tends to overlook the differences between samples, leading to variations in data projection after dimensionality reduction. To address this issue, this article incorporates the Pearson correlation coefficient for feature selection during data preprocessing.

The accounting fraud dataset for listed companies includes numerous indicators, such as company size, asset–liability ratio, current ratio, and quick ratio. However, the correlation between these indicators can negatively impact the accuracy of model training, resulting in poor discrimination and a lack of generalization. To mitigate this, the Pearson correlation coefficient method is used to calculate the correlation between multiple variables, as determined by the following formula: (2) ρ=CovX,YVarXVarY,

where ρ is the correlation between variables X and Y.

Rationale for model selection

In corporate financial accounting fraud detection, traditional methods like neural networks, logistic regression, and support vector machines are widely used. However, these models often fall short in delivering high accuracy due to the complex and evolving nature of fraud patterns. To overcome these challenges, this paper introduces an advanced approach that integrates CNN and LSTM networks. CNNs are effective at capturing spatial patterns and performing feature extraction, while LSTMs excel at handling temporal dependencies and sequence data, making this hybrid approach ideal for uncovering intricate fraud patterns in financial data. This hybrid approach has also demonstrated promising results in various other applications. For example, in time series anomaly detection, CNNs extract key features while LSTMs handle sequential dependencies, leading to effective irregularity detection. In healthcare, this combination has been used to analyze both spatial and temporal trends for predictive diagnosis (Nguyen et al., 2021), and in transportation systems, it has proven valuable for identifying fraudulent trips based on trajectory data (Ding et al., 2021; He et al., 2023). CNN-LSTM models have also shown significant potential in biological sequence data analysis (Le et al., 2019). For instance, they have been applied to predict protein structures and interactions, as well as to analyze DNA and RNA sequences by capturing complex spatial motifs and temporal dependencies inherent in biological data (Nguyen et al., 2019). These diverse applications highlight the robustness and adaptability of the CNN-LSTM hybrid model, making it highly suited to tackling the intricate and dynamic patterns found in financial fraud detection and beyond.

Moreover, we incorporate an attention mechanism to allow the model to prioritize the most relevant features, improving its ability to detect subtle anomalies that signal fraud. To maximize performance, we employ the SSA for parameter optimization, ensuring that the CNN-LSTM model operates at optimal efficiency. This method is specifically designed to overcome the limitations of traditional models and achieve higher detection accuracy for corporate accounting fraud.

Our proposed fraud detection framework follows a four-step process: data preprocessing, feature extraction from key indicators, internal change pattern identification, and hyperparameter optimization. We present the SSA-CNN-LSTM-Attention recognition model, which extracts and analyzes fraud-related characteristics and internal patterns in accounting data. By leveraging the strengths of neural networks and intelligent optimization algorithms, our approach aims to significantly improve detection accuracy. Figure 1 illustrates the architecture of our model, with further details provided in the subsequent sections.

Figure 1 Architecture of the proposed CNN-LSTM-Attention model.

Convolutional neural network

CNNs are used at processing multi-dimensional data and effectively extracting features from time series. Given that different indicators carry varying levels of importance, we developed a CNN-LSTM-Attention model to achieve a high-precision accounting fraud identification and classification system.

In our approach, the CNN is used for local feature extraction. It processes the input data through convolutional layers, refines the features using pooling layers, and then integrates the output via fully connected layers. The structure consists of convolutional layers, pooling layers, and fully connected layers. When data is fed into the model, the local blocks are convolved using a sliding window based on a predefined convolution size, which allows the model to capture the feature weights of the local regions. Pooling is then applied to eliminate features with smaller weights, removing redundant information and enhancing recognition accuracy.

However, during the convolutional feature extraction process, the model may overlook the correlation between features. Since the accounting fraud data of listed companies exhibits time series properties, understanding the dependencies between data points is crucial. To address this, we incorporated an LSTM network to extract and analyze these time series features.

The convolutional layer extracts features from the input data. During training, the network adjusts the kernel weights of the convolutional layer. This approach offers better storage efficiency and faster learning rates compared to other neural network models. Stacking multiple convolutional layers allows for the extraction of more complex features from the input data. The convolution kernel processes data by sliding from top to bottom and left to right, performing point-wise multiplication.

Following the convolutional layer is the pooling layer, which downsamples the data to re-extract features and simplify the network’s structure. This process helps to reduce computational complexity and retain essential information.

After the convolutional and pooling layers, the fully connected layer integrates the data. It connects to the neurons of the upper layer, converting the incoming data into a flat array. This array serves as the input for the fully connected layer, preserving a significant amount of useful information during the transformation.

In our method, the CNN architecture includes two convolutional layers, 2 max pooling layers, and a fully connected layer. Convolution is performed using 1-D operations, with the ReLU activation function applied in the convolutional layers and the Sigmoid function used in the fully connected layer. The CNN extracts Hc, which represents the feature vector: (3) C1=ReLUHt⊗W1+b1,P1= maxC1+b2,C2=ReLUP1⊗W2+b3,P2= maxC2+b4,Hc=SigmoidP2⊗W3+b5,

where C1 and C2 denote the outputs of the two convolutional layers, while P1 and P2 denote the outputs of the two pooling layers. W1, W2, and W3 are the weight matrices, and b1, b2, b3, b4, and b5 represent the biases. The symbol ⊗ indicates the convolution operation, and max refers to the maximum function.

LSTM neural network

The LSTM network, introduced by Hochreiter & Schmidhuber (1997), addresses the issues of vanishing and exploding gradients in traditional neural networks by incorporating memory cells and gating mechanisms. This innovation enables the network to retain long-term information and effectively handle dependencies and historical data in time series.

The LSTM network model is composed of a series of time series modules, each featuring an input gate, a forget gate, an output gate, and a memory unit. The input gate controls the information flow into the memory unit, the forget gate manages the retention of historical data, and the output gate regulates the information output. The forget gate updates the memory by evaluating the previous hidden state and the current input, determining how much of the past information should influence the current state. The input gate decides the amount of new information to store, using a sigmoid function to output values between 0 and 1; a value of 0 means no update is made, while 1 indicates that the information should be updated. The output gate determines how much of the information from the memory unit is passed to the next layer, also using a sigmoid function to output either 0 (no information passed) or 1 (information passed). This gating mechanism allows LSTM networks to selectively manage the flow of information, enabling the network to preserve and process relevant data while discarding irrelevant information.

The LSTM network can be expressed by the following formulas: (4) X=Xtht−1,ft=δWf⋅X+bf,it=δWi⋅X+bi,Ot=δWo⋅X+bo,ct=ft∘ct−1+it∘ tanhWc⋅X+bc,ht=Ot∘ tanhct,

where Xt denotes the input at time t; ht represents the hidden state at time t; Wf, Wi, Wo, and Wc are the weight matrices for the forget gate, input gate, output gate, and memory cell, respectively; bf, bi, bo, and bc are the corresponding biases for these gates and memory cell; ∘ denotes the dot product calculation; and δ represents the activation functions used in the network.

We employed the Adam algorithm to update the network weights. This algorithm iteratively adjusts the weights based on training samples, utilizing the second-order moment of the gradients to adaptively compute the learning rate for the parameters.

Attention mechanism

The attention mechanism is a model that mimics the way the human brain focuses on specific parts of information. In this mechanism, while input and output data are connected with varying weights in the hidden layer, the encoding output is incorporated into the attention range. The attention mechanism assigns different weights to input data features, amplifying certain information. This means that the concentrated information is received more sensitively and processed more quickly. In scenarios where LSTM takes too long to process data, its ability to retain information diminishes. By integrating the attention mechanism, which emulates how the human brain processes large volumes of information, the network’s information processing capabilities are enhanced. The attention mechanism alleviates the Encoder-Decoder model’s limitation on handling fixed-length vector data by calculating similarities. During this calculation, the input that is more similar to the target output receives a higher attention weight. For different outputs (y), the attention mechanism computes the similarity between each input (x) and the output (y), leading to different attention weights for each input. These weights are then used in a weighted summation, increasing the model’s flexibility.

The output vector from the LSTM layer is fed into the attention layer, where the probabilities corresponding to the LSTM output vector are calculated. The weight parameter matrix is then continuously updated as follows. (5) et=μtanhwht+b,at= expet ∑j=1tej,st= ∑t=1iatht,

where et represents the attention probability distribution μ at time t; w denotes the weight coefficients; b represents the bias coefficients; at is the attention weight; and st is the output of the Attention layer at time t.

Model optimization using sparrow search

SSA is a recent advancement in swarm intelligence. It categorizes the sparrow population into two groups: discoverers and joiners. A fitness function is employed to calculate each sparrow’s fitness value, which in turn facilitates the exchange of roles and positions within the group. Unlike genetic algorithms, particle swarm optimization, or ant colony algorithms, SSA effectively mitigates the risk of converging prematurely to a suboptimal solution.

Let X represent the randomly initialized sparrow population, where (6) X=x11x12…x1dx21x22…x2d⋮⋮…⋮xn1xn2⋯xnd,

d is the dimensionality of the population, and n is the number of sparrows. The fitness of each sparrow, F, is then defined as follows: (7) Fx=fx11x12⋯x1dfx21x22⋯x2d⋮fxn1xn2⋯xnd.

We have (8) Xi,jt+1=Xi,j⋅exp −1α⋅itermax,ifR2<STXi,j+Q⋅L,otherwise

where t represents the current iteration number, itermax is the maximum number of iterations, and Xi,j is the position of the ith sparrow in the jth dimension. Here, α, R2 ∈ (0, 1] and ST ∈ (0.5, 1] denote the early warning value and safety value, respectively. Q represents random numbers following a Gaussian distribution, and L is a 1 × d matrix with all elements equal to 1.

When a discoverer within the sparrow population detects a predator, it will immediately raise an alarm. If the alarm value exceeds the safety threshold, the discoverer will update its location. Otherwise, it will continue searching within the space: (9) Xi,jt+1=Q⋅exp(Xworst−Xi.jti2),ifi>n2Xpt+1+Xi,j−Xpt+1⋅A+⋅L,otherwise

where Xp represents the optimal position currently held by the discoverer, Xworst denotes the current global worst position, and A is a matrix where each element is randomly assigned a value of either 1 or −1, with A+ = AT(AAT)−1.

During foraging, the joiner observes the finder. Once the finder locates food, the joiners immediately compete for it. If a joiner wins, it takes over the finder’s food; otherwise, it continues searching the space for food. The joiner updates its position according to the formula. If a joiner’s fitness value is low and it fails to obtain food, it will exchange positions to increase its chances of finding more food: (10) Xi,jt+1=Xbestt +β⋅Xi,jt−Xbestt,iffi<fgXi,jt+K⋅Xi,jt−X worsttfi−fw+ɛ,otherwise

where Xbestt is the global optimal position at the current iteration, β is a step control parameter that follows a N(0, 1) distribution, and K ∈ [ − 1, 1]. fi represents the fitness value of the ith sparrow, while fg and fw denote the global optimal and worst fitness values, respectively. The term ɛ is an adjustment factor introduced to prevent the denominator from becoming zero and causing an undefined or infinite result.

The objective function is (11) MAE=1n∑i=1nyi−y ˆi,

where n is the number of samples, yi is the true value, and y ˆi is the predicted value.

The randomly initialized discoverers and joiners compete for food and continuously update their positions, searching for the sparrow with the highest global fitness value as the global optimum. This process continues until the maximum number of iterations is reached. As a result, the SSA is utilized to optimize the hyperparameters of the CNN-LSTM-Attention model. The process of SSA optimizing the hyperparameters of the CNN-LSTM-Attention model is illustrated in Fig. 2.

Figure 2 Hyperparameter tuning using the Sparrow Search Algorithm.

Assessment metrics

In the financial accounting of listed companies, accounting fraud has long been a significant concern. To effectively identify potential accounting fraud, discriminant detection technology has become an essential tool. When assessing the performance of a discriminant detection algorithm, several indicators are typically used to quantify its effectiveness, including detection rate (DR), false acceptance rate (FAR), accuracry (ACC), and F1-score. DR measures the percentage of abnormal data correctly identified as outliers. In the context of financial accounting, it reflects the successful identification of potential accounting fraud. The formula for calculating DR is: DR=TPTP+FN,

where TP (true positives) and FN (false negatives) represent the number of correctly identified fraudulent cases and the number of missed fraudulent cases, respectively. A high DR indicates that the model is effective in detecting potential fraud.

FAR represents the percentage of normal data values incorrectly classified as outliers. In the context of financial accounting detection, FAR indicates the proportion of legitimate situations mistakenly flagged as potential fraud. The formula for calculating FAR is: FAR=FPFP+TN,

where FP (false positives) and TN (true negatives) correspond to the number of incorrectly identified fraudulent cases and the number of correctly identified non-fraudulent cases, respectively. A lower FAR indicates that the model performs better in minimizing false alarms.

ACC refers to the proportion of correct detections, representing the model’s ability to accurately identify both positive and negative samples. The formula for calculating ACC is: ACC=TP+TNTP+TN+FP+FN.

The F1-score is a metric that combines both precision and recall to provide a comprehensive measure of a model’s performance. The formula for calculating the F1-score is: F1-score=2×Precision×RecallPrecision+Recall=TPTP+12FP+FN,

where precision represents the proportion of positive predictions that are correctly identified as positive, and recall represents the proportion of actual positive cases that are correctly identified. A higher F1-score indicates a better balance between precision and recall.

Experimental Results and Discussion

The experiments were conducted on a system with a 64-bit Windows 11 operating system, 32GB of RAM, 8GB of GPU memory, and a Core™ i7-13620H processor. This study compares the proposed SSA-optimized CNN-LSTM-Attention model with five other prediction models, analyzing accounting fraud detection for listed companies in 2020 and 2021. Additionally, it includes a longitudinal study of accounting fraud detection spanning seven years, from 2015 to 2021, to further evaluate the performance of the SSA-optimized CNN-LSTM-Attention model.

The objective function curve across various iterations is illustrated in Fig. 3. The figure illustrates that as the iterations increase, the objective function curve decreases and ultimately converges, achieving nearly 60 iterations, with the objective function value approximating 0.51.

Figure 3 Evaluation of accounting fraud detection results across different algorithms.

Figure 4 Evaluation of accounting fraud detection results across different algorithms.

Different evaluation metrics are calculated from the sparse confusion matrix, including DR, ACC, FAR, and F1-score, to evaluate the effectiveness of various algorithms in identifying accounting fraud in listed companies. The results are presented in Fig. 4. As shown in Fig. 4, the SSA-CNN-LSTM-Attention model demonstrates superior performance, higher discrimination accuracy, and more precise results compared to the other models.

Figure 5 illustrates DR of four methods with varying values of k, a, and multi-variable data. As shown, the CNN-LSTM-Attention model consistently achieves higher DR compared to CNN, LSTM, and CNN-LSTM across different k and a values. Notably, when k = 102 and a = 0.005, CNN-LSTM-Attention’s DR is approximately 3.21% higher than that of CNN-LSTM.

Figure 5 DR for four methods with multivariate data and different k and a.

Figure 6 presents FAR for the same methods under different k and a values. It is evident that CNN-LSTM-Attention outperforms CNN, LSTM, and CNN-LSTM in terms of FAR, with a reduction of nearly 4.128% compared to other algorithms.

Figure 6 FAR for four methods with multivariate data and different k and a.

The results of accounting fraud detection vary with different numbers of hidden layer nodes and learning rates. This paper uses the Sparrow algorithm to optimize CNN-LSTM-Attention, demonstrating that the optimal parameters are a hidden layer count of 102 and a learning rate of 0.0056. These parameters yield the highest accuracy, closely aligning with experimental results and confirming the effectiveness of the proposed algorithm.

Limitations and Future Work

While the proposed method demonstrates promising results in predicting corporate accounting fraud, there are several limitations that need to be addressed. One primary limitation is the dependence on the quality and quantity of the training data. Fraudulent transactions are typically rare and often involve complex, evolving patterns that are not fully captured in available datasets. This scarcity and imbalance of data can limit the model’s ability to generalize effectively to unseen cases, leading to potential overfitting or poor performance on real-world data. Another limitation is the computational complexity of the proposed method. The integration of CNN and LSTM networks, combined with an attention mechanism and SSA for parameter optimization, requires significant computational resources for training, especially on large datasets. This could pose challenges for scalability and deployment in real-time applications, where rapid detection of fraud is critical. Furthermore, the parameter optimization process using SSA, while effective, can be time-consuming, potentially delaying the availability of updated models as new data becomes available. This could hinder the method’s adaptability to changing fraud patterns, reducing its long-term effectiveness.

For future work, exploring strategies to mitigate data scarcity and imbalance, such as using data augmentation techniques or synthetic data generation, could enhance the model’s robustness and generalization capabilities. Additionally, incorporating semi-supervised or unsupervised learning approaches might help the model detect novel fraud patterns without relying solely on labeled data.

Conclusions

We utilized the SSA algorithm to optimize the parameters of the CNN-LSTM-Attention model and tested it with various proportions of test sets. Simulation experiments were conducted, evaluating the results using DR, FAR, ACC, and F1-score. The findings indicate that the SSA-optimized CNN-LSTM-Attention model outperforms other algorithms, demonstrating higher discrimination accuracy. Additional simulation experiments with different numbers of hidden layers and learning rates were performed. The results confirm that the parameters optimized by the sparrow algorithm within the same search space closely match the experimental results, validating the effectiveness of the proposed algorithm.

Supplemental Information

Supplemental Information 1 The code used in the study

Supplemental Information 2 Instructions for downloading the data

Supplemental Information 3 Instructions for reproducing the results

Supplemental Information 4 Discriminant variables

Supplemental Information 5 Dictionary for the discriminant variables

Supplemental Information 6 Matching data

Supplemental Information 7 Dictionary for the matching data

Additional Information and Declarations

Competing Interests

Author Contributions

Data Availability

The authors declare there are no competing interests.

Peifeng Wu conceived and designed the experiments, performed the experiments, analyzed the data, performed the computation work, prepared figures and/or tables, authored or reviewed drafts of the article, and approved the final draft.

Yaqiang Chen conceived and designed the experiments, performed the experiments, analyzed the data, prepared figures and/or tables, authored or reviewed drafts of the article, and approved the final draft.

The following information was supplied regarding data availability:

The original dataset is available at the CSMAR website (https://data.csmar.com).

The downloaded dataset, the accompanying code and detailed instructions for downloading the data are available in the Supplemental Files.

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
