# Peer review of "Enhanced detection of accounting fraud using a CNN-LSTM-Attention model optimized by Sparrow search"

_PeerJ Computer Science, doi:10.7717/peerj-cs.2532_

## Round 0.1 · original submission · Major Revisions

After careful consideration and review, we have decided that major revisions are needed before it can be further considered for publication.

Attached are the detailed comments and suggestions from the two reviewers. Please revise the manuscript accordingly and address all the reviewers' comments in a point-by-point response.

Reviewer 1 ·

Basic reporting

The manuscript is well-written and uses clear technical language. The introduction and background sections give good context for the study, but it could benefit from including more relevant previous research. The figures are high-quality, presented in vector format, and are well-integrated into the manuscript. They relate well to the content and are clearly described and labeled.
The code and data were distributed as a supplemental document, making it easily accessible and reproducible.

Experimental design

The importance of the introduction could be further emphasized by adding more references to support its significance.
The paper employs the sparrow search algorithm. Please include the objective function iteration graph for this algorithm and provide a more detailed explanation of the optimization results.
The authors could add another set of experiments by including data from 2015 for algorithm testing and presenting the algorithm's performance in a table format.

Validity of the findings

It seems that the experiments and evaluations performed satisfactorily. However, conducting an additional experiment, as suggested above, could further validate the technical reliability of the results.
The conclusions are well-articulated, directly address the original research question, and are confined to what is supported by the experimental findings. However, the conclusion or the text just before it should also discuss limitations and/or future directions.

Additional comments

Equation 8 has some shortcomings, as it does not account for all possible scenarios. While it addresses the conditions "if R2<ST" and "if R2>ST," it fails to specify what occurs when "R2=ST." It is advisable to include an else statement for clarity.
References are not properly cited; for instance, line 147 lacks a citation for the source of the LSTM network.
While the comments above can be addressed, the manuscript is not yet ready for acceptance and requires revision.

Reviewer 2 ·

Basic reporting

The manuscript is well-structured and written in clear, formal English. The literature review is relatively comprehensive and offers a thorough overview of the topic. The tables and figures are well-designed, effectively illustrating the research findings. To enhance the quality of the manuscript, additional visualizations could be included, and specific points of clarification should be addressed, as detailed below.

Experimental design

+ To improve clarity, it would be helpful to include a flowchart illustrating the optimization logic of the Sparrow search algorithm used in the paper. This will provide readers with a visual understanding of the algorithm's process and how it was applied to optimize the model.
+ The objective function optimized by the sparrow search algorithm is not sufficiently explained. Please provide further details to clarify how the objective function is defined and optimized.
+ Consider plotting the prediction accuracy curves of different algorithms over multiple years: This would provide additional validation of the algorithm's feasibility.

Validity of the findings

+ The manuscript introduces a neural network-based deep learning model for identifying and detecting financial fraud in companies. Overall, the findings appear to be valid.
+ Please provide a more detailed quantitative analysis of Figure 2, as it currently only includes qualitative descriptions without supporting numerical data.
+ It is recommended to remove the F1_SCORE index calculation for adjustment and instead summarize the performance evaluation of all four indicators together to enhance readability.

Additional comments

+ Equation 10 is not described in a standardized manner: For example, the cases "if fi < fg" and "if fi = fg" are described, but there is no mention of the scenario when "fi > fg".
+ Equations 8, 9, and 10 are not properly formatted: it is recommended to leave a space after the if statements.

---

## Round 0.2 · accepted · Accept

Both reviewers have confirmed that the authors have addressed all previous comments thoroughly, and, based on their recommendations, I am pleased with the current version and find it ready for publication.

Reviewer 1 ·

Basic reporting

The manuscript is now even clearer and well-structured, with additional references in the introduction that strengthen the context and relevance of the study. The figures continue to be of high quality and are effectively integrated. The supplemental code and data have been maintained for reproducibility, further enhancing the paper’s transparency.

Experimental design

The authors have successfully incorporated an objective function iteration graph for the sparrow search algorithm, providing a detailed explanation of the optimization results.

Validity of the findings

The expanded experimental setup and additional analyses lend further credibility to the results. The discussion on limitations and future research directions, added to the conclusion, demonstrates a balanced and forward-looking perspective. The findings are now well-supported by a robust methodology and comprehensive discussion.

Additional comments

Overall, the manuscript has improved significantly and now meets the standards required for publication. I recommend it for acceptance.

Reviewer 2 ·

Basic reporting

The revised manuscript is well-structured and clearly written, with a comprehensive literature review. I have no further comments.

Experimental design

The authors have successfully included a flowchart for the sparrow search algorithm, providing a clear visual overview of the optimization process. The additional details on the objective function clarify its definition and optimization approach, making this section more informative.

Validity of the findings

No further comments.

Additional comments

No further comments.